# Electrographic Seizures in Neonates with a High Risk of Encephalopathy

**DOI:** 10.3390/children9060770

**Published:** 2022-05-24

**Authors:** Wan-Hsuan Chen, Oi-Wa Chan, Jainn-Jim Lin, Ming-Chou Chiang, Shao-Hsuan Hsia, Huei-Shyong Wang, En-Pei Lee, Yi-Shan Wang, Cheng-Yen Kuo, Kuang-Lin Lin

**Affiliations:** 1Department of Pediatrics, Chiayi Chang Gung Memorial Hospital and Chang Gung University College of Medicine, Chiayi 613, Taiwan; wererabbit122@hotmail.com; 2Division of Pediatric Critical Care and Pediatric Neurocritical Care Center, Chang Gung Children’s Hospital and Chang Gung Memorial Hospital, Chang Gung University College of Medicine, Taoyuan 333, Taiwan; ai3333@cgmh.org.tw (O.-W.C.); tw1picu@gmail.com (S.-H.H.); pilichrislnp@gmail.com (E.-P.L.); 3Division of Pediatric Neurology, Chang Gung Children’s Hospital and Chang Gung Memorial Hospital, Chang Gung University College of Medicine, Taoyuan 333, Taiwan; wanghs444@cgmh.org.tw (H.-S.W.); hermitage513@gmail.com (Y.-S.W.); cykuo2000@hotmail.com (C.-Y.K.); lincgh@cgmh.org.tw (K.-L.L.); 4Division of Neonatology, Chang Gung Children’s Hospital and Chang Gung Memorial Hospital, Chang Gung University College of Medicine, Taoyuan 333, Taiwan; newborntw@gmail.com; 5Study Group for Intensive and Integrated Care of Pediatric Central Nervous System, Department of Pediatrics (iCNS Study Group), Chang Gung Memorial Hospital, Taoyuan 333, Taiwan; bread86@cgmh.org.tw

**Keywords:** continuous, electroencephalographic monitoring, electrographic seizure, neonate, a high risk of encephalopathy

## Abstract

Background: Neonatal encephalopathy is caused by a wide variety of acute brain insults in newborns and presents with a spectrum of neurologic dysfunction, such as consciousness disturbance, seizures, and coma. The increased excitability in the neonatal brain appears to be highly susceptible to seizures after a variety of insults, and seizures may be the first clinical sign of a serious neurologic disorder. Subtle seizures are common in the neonatal period, and abnormal clinical paroxysmal events may raise the suspicion of neonatal seizures. Continuous video electroencephalographic (EEG) monitoring is the gold standard for the diagnosis of neonatal seizures. The aim of this study was to identify the prevalence of electrographic seizures and the impact of monitoring in neonates with a high risk of encephalopathy. Methods: We conducted this prospective cohort study in a tertiary neonatal intensive care unit over a 4-year period. Neonates with a high risk of encephalopathy who were receiving continuous video EEG monitoring were eligible. The patients were divided into 2 groups: (1) acute neonatal encephalopathy (ANE) and (2) other high-risk encephalopathy conditions (OHRs). The neonates’ demographic characteristics, etiologies, EEG background feature, presence of electrographic seizures and the impact of monitoring were analyzed. Results: A total of 71 neonates with a high risk of encephalopathy who received continuous video EEG monitoring were enrolled. In this consecutive cohort, 42 (59.2%) were monitored for ANE and 29 (40.8%) were monitored for OHRs. At the time of starting EEG monitoring, 54 (76.1%) of the neonates were term infants. The median gestational age at monitoring was 39 weeks (interquartile range, 37–41 weeks). The median total EEG monitoring duration was 64.7 h (interquartile range, 22.2–72.4 h). Electrographic seizures were captured in 25 of the 71 (35.2%) neonates, of whom 20 (80%) had electrographic-only seizures without clinical correlation. Furthermore, of these 20 neonates, 13 (65%) developed electrographic status epilepticus. Electrographic seizures were most commonly found in the ANE group (17, 40.5%) than in the OHRs group (8, 27.6%) (*p* = 0.013). Besides, normal/mild abnormality and inactive EEG background were less electrographic seizure than moderate and major abnormality EEG background (2 of 30, 6.7% vs. 23 of 41, 56.1%, *p* < 0.001). Finally, continuous video EEG monitoring excluded the diagnosis of electrographic seizures in two-thirds of the monitored neonates who had paroxysmal events mimicking seizures and led to a change in clinical management in 39.4% of the neonates. Conclusions: Our findings showed that monitoring could accurately detect seizures, and that it could be used to guide seizure medication management. Therefore, continuous video EEG monitoring has important clinical management implications in neonates with a high risk of encephalopathy.

## 1. Introduction

Neonatal encephalopathy is caused by a wide variety of acute brain insults in newborns and presents with a spectrum of neurologic dysfunction, such as consciousness disturbance, seizures, and coma [1,2]. Seizures may be the first clinical sign of a serious neurologic disorder and are the neurological emergencies that need immediate management. The overall incidence of acute symptomatic seizures in all ages is 16 to 39 per 100,000 per year, and acute symptomatic seizures predominate in those less than 1 year of age and, to a lesser extent, in the elderly [3,4]. The incidence of neonatal seizures is estimated to be 200 to 500 cases per 100,000 term infants, approximately 5000 per 100,000 premature infants [5], and 55 to 100 per 100,000 patients older than age 60 years [6].

The developing brain was more hyperexcitable than the adult brain. In the developing brain, the excitatory glutamatergic neurons are overabundant and GABAergic neurons exert a paradoxical excitatory action. The Na-K-2Cl cotransporter isoform 1 (NKCC1) and the K-Cl cotransporter isoform 2 (KCC2) play an important role of GABA receptors’ function by regulating intracellular chloride concentration. The fundamental role of NKCC1 in establishing excitatory GABAergic neurotransmission in the neonate and KCC2 is the opposite [7].

The increased excitability in the neonatal brain appears to be highly susceptible to seizures, and epileptic focus may result from a variety of acute brain insults [8]. In epileptic focus, the neurons are bursters with the paroxysmal depolarization shift (PDS) [8]. There are many mechanisms involved in the seizure propagation. Guan et al. reported that Kv1 channels differentially regulate action potentials and repetitive firing in two pyramidal neuron types, and blocking of Kv1 channels can help in converting single action potential toward bursting [9]. Chen et al. reported that small-conductance Ca(2+)-activated K(+) (SK or K(Ca)2) channels were shown to become activated during repetitive firing, causing early spike frequency adaptation [10]. However, GABAergic activity is involved in stopping seizure propagation [8]. These suggested that multiple cellular level mechanisms govern the propagation and termination of epileptic activity.

Seizures are common in these neonates; however, most are subclinical or electrographic-only [11,12]. On the other hand, an epileptic seizure is a transient occurrence with signs or symptoms due to abnormal excessive and synchronous neuronal activity in the brain. Pathak et al. proposed that Na+ channels and calcium-activated potassium current (IK(Ca)) might have broader significance on membrane potential oscillations, burst firing, and seizure activity [13]. Therefore, abnormal paroxysmal events, such as abrupt, repetitive, or abnormal appearing movements, atypical behavior or unprovoked episodes of autonomic dysfunction, may be the clinical manifestation of neonatal seizures [14,15].

Monitoring seizure activity with clinical observations alone in neonates with a high risk of encephalopathy can lead to underestimation, as only one-third of all seizures present with clinical features [16,17]. Increasing clinical studies show that neonatal seizures, especially status epilepticus, are at a high risk of poor neurodevelopmental outcomes, such as development delay, intellectual disability and epilepsy [18,19].

Seizures may also be overdiagnosed by clinical observation, and clinical paroxysmal events may not show corresponding electrographic evidence of seizures [17,20,21]. In a study of critically ill neonates, 13% of the neonates who were monitored for paroxysmal events mimicking seizures showed no evidence of electrographic seizures [20]. Hence, determining the optimal method to identify neonatal seizures is important in neonates with a high risk of encephalopathy. Continuous EEG monitoring is now recommended for the seizures detection and optimizing treatment for patients who truly need medication [12]. The aims of the current study were to identify the prevalence of electrographic seizures and the impact of monitoring in neonates with a high risk of encephalopathy.

## 2. Methods

### 2.1. Patient Population

Neonates admitted to the neonatal intensive care unit (NICU) of Chang Gung Memorial Hospital from January 2017 to December 2020 were prospectively enrolled into this study. We included neonates with a high risk of encephalopathy who were candidates for continuous video EEG monitoring based on guidelines published by the American Clinical Neurophysiology Society (ACNS) in 2011, including: (1) acute neonatal encephalopathy (ANE), and (2) other high-risk encephalopathy conditions (OHRs) [12]. ANE included infants suspected of having hypoxic-ischemic injury, such as perinatal asphyxia or following cardiopulmonary resuscitation. OHRs included any of the following conditions: perinatal stroke, central nervous system infections, genetic diseases/inborn errors of metabolism/metabolic disorders, prematurity with additional risk factors (such as intraventricular hemorrhage), and others (congenital malformations and cardiac or pulmonary risk factors, such as Tetralogy of Fallot and extra-corporeal membrane oxygenation). We excluded patients who: (1) were monitored for a differential diagnosis of paroxysmal events mimicking seizures; (2) were monitored for seizure recurrence during or after weaning from anti-seizure medications; (3) were aged ≥48 weeks postmenstrual age; and (4) had poor quality video-EEG recordings. Demographic characteristics and etiologies were extracted from the hospital’s database and chart review. This study was approved by the Ethics Committee of Chang Gung Medical Hospital (protocol code: 201600360B0; date of approval: 18 April 2016; protocol code: 201901000B0; date of approval: 17 July 2019). Informed consent was obtained from all subjects involved in the study.

### 2.2. Continuous Video EEG Monitoring Protocol

A standardized continuous video EEG monitoring protocol for neonates in NICU was initiated at Chang Gung Children’s Hospital in October 2016, according to ACNS consensus-based guidelines proposed in 2011 [12]. Bedside 16-channel continuous EEG monitoring was performed using a Nicolet Monitor (Natus Neuro, Middleton, WI, USA) video-EEG system and a modified international 10–20 system for neonatal montage, including eleven scalp electrodes (FP2, C4, T4, O2, FP1, C3, T3, O1, Fz, Cz, Pz), was used with additional extracerebral leads for respiratory and electrocardiogram recordings, which is necessary to accurately evaluate behavioral state and exclude extracerebral artifacts [20]. Because all neonates were with a high risk of acute encephalopathy, they were in fasting status during the recording. If a bedside observer alerted a clinically relevant event, the staff pressed a push button and documented it on a flow sheet.

Monitoring duration and discontinuation of EEG monitoring was adapted from the clinical pathway for children who require EEG monitoring in the Children’s Hospital of Philadelphia (EEG Monitoring Clinical Pathway-NICU-Children’s Hospital of Philadelphia (chop.edu)). In general, the duration of EEG monitoring is determined by the clinical context. EEG monitoring is generally terminated after 24–48 h if no electrographic seizures are identified, and no major changes have occurred in the patient’s condition. If electrographic seizures occur, then EEG monitoring is continued until the patient has been seizure-free for at least 24 h. EEG monitoring may extend longer in the following scenarios: (1) ongoing risk for electrographic seizures; (2) evolving acute neurologic brain injury present; (3) anti-seizure or sedative medications are being weaned/adjusted. EEG monitoring may be terminated more quickly if: (1) patient is rapidly improving; (2) EEG background suggests a very low risk for electrographic seizures; (3) no additional seizure management clinically indicated [22].

EEG recording formats were stored, not only the EEG signal data but also the technologist’s comments and push-button event codes.

### 2.3. EEG Interpretation

Continuous video EEG data with patient’s clinical information were analyzed visually by an experienced neurophysiologist, based on the double banana montage with an analog bandwidth of 1–70 Hz and a notch filter of 60 Hz. Our EEG systems also have quantitative display functions, including amplitude-integrated EEG (aEEG), which allow detection of long-term changes in EEG signals at a glance and thus are useful for screening purposes.

Each EEG file was interpreted using standardized terminology of the American Clinical Neurophysiology Society (ACNS) for neonates [23]. In brief, EEG background feature and presence of electrographic seizures were reviewed. EEG background was categorized as (1) normal/mild abnormalities: normal pattern for gestational age (GA), including mild asymmetries, mild voltage depression, (2) moderate abnormalities: discontinuous activity with interburst intervals (IBI) too long for GA or clear asymmetry and asynchrony, (3) major abnormalities: severe discontinuity in EEG for GA or burst suppression pattern, and (4) inactive EEG: background activity <10 μV [24]. Electrographic seizures were defined as an abnormal abrupt background activity with evolution in frequency, morphology, and spatial distribution, with a duration of 10 s on EEG [23,25]. Distinct seizures were separated by a seizure-free interval of at least 10 s. The electrographic seizures were divided into two categories according to clinical manifestation: electroclinical seizures and electrographic-only seizures. Electroclinical seizures, also called convulsive or clinically evident seizures, can present with any duration. In contrast, electrographic-only seizures, also called non-convulsive or subclinical seizures, are often defined as lasting for 10 s or more [23,25]. Electrographic status epilepticus was defined as continuous electrographic seizures lasting at least 30 min or repeated electrographic seizures totaling more than 30 min in any 1 h period [23,25]. Each EEG file was also annotated for the timings of seizure initiation, propagation, and termination using the Nicolet Reader software (Natus Neuro, Middleton, WI, USA). We only calculated the seizure frequency, and we do not calculate the duration of every seizure episode. Therefore, we just present the seizure frequency. Besides, we submitted videos of electrographic-only and electroclinical seizures in the supplement file to demonstrate the important role of continuous video EEG monitoring (Appendix A, Appendix A and Figure 1).

### 2.4. Seizure Outcome

Seizure outcomes were evaluated at 12 months of age and were classified into three groups: (1) intractable epilepsy; (2) favorable outcome; and (3) seizure-free without the use of AEDs. Intractable epilepsy was defined as more than two seizures per month in patients receiving two or more antiepileptic drug treatments. Favorable outcome was defined as either seizure-free or fewer than two seizure episodes per month after treatment [26].

### 2.5. Statistical Analysis

The patients were divided into 2 groups: those with ANE and those with OHRs. Because of the small number of neonates in each of the OHR categories, they were analyzed together. The patient characteristics, EEG finding, treatment, and seizure outcomes in 2 groups are shown as number with percentage, or median with interquartile range (IQR). The Chi-squared test or Fisher’s exact test for categorical variables was used to examine differences between groups, and the Mann–Whitney U test was used to compare medians. Statistical analyses were performed using SPSS version 23 (IBM, Inc., Chicago, IL, USA). In all instances, a *p* value < 0.05 was considered to indicate a statistically significant difference.

## 3. Results

### 3.1. Patient Profile

Seventy-one neonates with a high risk of encephalopathy were enrolled over the 4-year observation period (Figure 1). In this consecutive cohort, 42 (59.2%) were monitored for ANE and 29 (40.8%) were monitored for OHRs. Seventy-nine infants were excluded, including 58 who were monitored for a differential diagnosis of abnormal paroxysmal events mimicking seizures, 15 who were monitored for seizure recurrence during or after weaning from anti-seizure medications, three who were aged ≥48 weeks postmenstrual age, and three who had poor quality recordings. Fifty-eight neonates who were monitored for a differential diagnosis of abnormal paroxysmal events mimicking seizures included 26 healthy term babies without a risk of encephalopathy, 15 prematurity without additional risk factors (such as intraventricular hemorrhage), and 17 neonates with previous history of acute encephalopathy with seizures and received the anticonvulsant before the continuous video EEG monitoring. 

Of the 71 enrolled neonates, 45 (63.4%) were boys, and at the time of initiating continuous EEG, 54 (76.1%) were term infants. The median gestational age at monitoring was 39 weeks (IQR 37–41 weeks). Although there is a sex difference between two groups, we examined the sex difference between two groups in the likelihood to have seizures. The likelihood ratio is 7.782, but there is no statistically significant difference (*p* = 0.549). There was also no statistical difference in gestational age at the initiation of monitoring. Besides, only 12 neonates had clinical symptoms or signs that were confirmed as clinical seizure by continuous video EEG monitoring. The most common seizure types were clonic seizures in the ANE group and clonic and subtle seizures in the OHRs group (Table 1).

The median total EEG monitoring duration was 64.7 h (IQR 22.2–72.4 h). The ANE group (71.6 h, IQR 64.9–78.4 h) was monitored for a significantly longer duration than the OHRs group (18.9 h, IQR 16.6–23.7 h) (*p* < 0.001). Finally, six (8.4%) neonates died before hospital discharge, including 4 (9.5%) in the ANE group and 2 (6.8%) in the OHRs group. Besides, 34 of the 65 (52.3%) neonates who survived were discharged with anti-epileptic drugs (AEDs). In terms of seizure outcome after 1-year follow-up in 63 neonates, 40 were seizure-free without the use of AEDs, 15 had favorable outcomes and 8 had intractable epilepsy (Figure 1). The characteristics of the 71 neonates who were monitored using continuous video EEG are summarized in Table 1.

### 3.2. Indications for Continuous EEG Monitoring and Findings

Electrographic seizures were captured in 25 of the 71 (35.2%) neonates, including five (20%) with electroclinical seizures, 11 (44%) with electrographic-only seizures, and nine (36%) with both electroclinical and electrographic-only seizures (Appendix A). Electrographic seizures were most commonly found in the ANE group (17, 40.5%) than in the OHRs group (8, 27.6%) (*p* = 0.013). Besides, of these 25 neonates, 20 (80%) had electrographic-only seizures without clinical correlation, and 13 (65%) had electrographic status epilepticus. Electrographic-only seizures were also most commonly found in the ANE group (14, 33.3%) than in the OHRs group (6, 20.6%), but there is no statistically significant difference (*p* = 0.244). Finally, the frequency of electrographic seizures was higher in the ANE group (median = 10, IQR 2–54.5) than in the OHRs group (median 5.5, IQR 1–11.25) (*p* = 0.051), Figure 2.

Most among the ANE group (33 of 42, 78.5%) were born at term. Therapeutic hypothermia was initiated in 22 (66.7%) of the 33 term neonates in the ANE group, and 21 completed the therapy (one neonate stopped therapy due to uncontrolled bleeding). Because hypothermia therapy only applied in the term neonate with moderate and severe HIE, it is rational that there are higher electrographic seizures in the hypothermia therapy group (12/23, 52.2%) than the without hypothermia therapy group (2/10, 20%) in HIE neonates, but there is no statistically significant difference (*p* = 0.131). 

Thirty-five of the 42 (83.3%) neonates in the ANE group had clinical paroxysmal events that were suspected of being seizures prior to EEG monitoring. Finally, 17 (40.5%) were found to have electrographic seizures, including 14 (82.3%) with electrographic-only seizures. In the OHRs group, 16 of the 29 (55.2%) neonates had clinical paroxysmal events that were suspected of being seizures prior to EEG monitoring. Lastly, 8 (27.6%) were found to have electrographic seizures, including 6 (75%) with electrographic-only seizures. Electrographic seizures were most often seen in the neonates with acute neonatal encephalopathy. Because AED was prescribed for clinical suspicion of neonatal seizures by clinical staff, there is a significantly higher occurrence of electrographic seizure in the prescribing AED group (25 of 51, 49%) than the non-prescribing AED group (0 of 20, 0%) (*p* < 0.001). The electrographic seizures rates classified by the indication for monitoring are summarized in Table 2.

### 3.3. Impact of Continuous Video-EEG on Clinical Management

In this study, 51 of the 71 (71.8%) neonates with clinically abnormal paroxysmal events which staff believed to be a clinical seizure received an AED during the continuous video-EEG monitoring. After review, a total of 303 push-button episodes were recorded in 35 neonates, and 110 of these 303 (36.3%) episodes were identified to be electroclinical seizures in the simultaneous video recording in 14 neonates. In addition, continuous video-EEG monitoring led to a change in clinical management in 39.4% of the neonates (28 of 71), including initiating AEDs (2, 2.8%), AED escalation (11, 15.5%), and AED discontinuation (15, 21.1%) (Figure 2).

In 28 neonates who had continuous video-EEG monitoring that led to a change in AED treatment, 24 neonates were followed up for 1 year. In initiating the AEDs group (*n* = 2), 1 had a favorable outcome and 1 was seizure-free without the use of AEDs. In the AED escalation group (*n* = 11), 4 were expired, 2 had intractable epilepsy, 3 had a favorable outcome and 2 were seizure-free without the use of AEDs. In the AED discontinuation group (*n* = 15), 1 had intractable epilepsy, 5 had a favorable outcome and 9 were seizure-free without the use of AEDs (Figure 2).

## 4. Discussion

Continuous video-EEG monitoring is recommended for the seizure detection, as well as seizure management in neonates with a high risk of encephalopathy. In this study, one-third of the neonates with a high risk of encephalopathy had electrographic seizures, and 80% of these neonates had electrographic-only seizures. Furthermore, continuous video-EEG monitoring excluded the diagnosis of electrographic seizures in two-thirds of the monitored neonates who had paroxysmal events mimicking seizures and led to a change in clinical management in 39.4% of the neonates. This finding supports the use of the ACNS guidelines in neonates with a high risk of encephalopathy.

The increased excitability in the neonatal brain appears to be highly susceptible to seizures and epileptic focus may result from a variety of acute brain insults [8]. In epileptic focus, the neurons are bursters with the paroxysmal depolarization shift (PDS) [8]. There are multiple mechanisms involved in the seizure propagation. Dysfunction of potassium (K+) channels, including Kv2 and KV7 and SK channels, may play crucial roles in the regulations of neuronal excitability, but there is still much to discover about the roles for these channels.

Neonates who are suspected of having or demonstrate acute brain injury, especially those with concomitant encephalopathy, are at a high risk of developing electrographic seizures. Previous studies of neonates with a high risk of encephalopathy have reported electrographic seizures in 33–65% of those with hypoxic-ischemic encephalopathy who underwent therapeutic hypothermia, 90% of those with stroke, and 85% of those with central nervous system infections [25,27,28,29,30,31,32]. However, this is not observed in this study, and this may be due to small case numbers in our study.

In addition, 14–43% of neonates with electrographic seizures have been reported to develop electrographic status epilepticus [25]. In a multicenter study of neonates with hypoxic ischemic encephalopathy, half of those had seizures during therapeutic hypothermia [27]. Nash et al. also reported that, of 41 newborns with hypoxic ischemic encephalopathy who underwent therapeutic hypothermia, one-third developed electrographic seizures and 10% developed status epilepticus [29]. These findings are similar to our study. Therefore, in neonates with a high risk of encephalopathy, particularly if they have encephalopathy or present with abnormal neurological examination findings, continuous video-EEG monitoring is necessary to identify electrographic seizures.

The gap between clinical diagnosis of neonatal seizures by staff observation and electrographic seizures has been described. Many infants with a high risk of encephalopathy have a variety of clinical paroxysmal events mimicking seizures, but do not have electrographic correlation. Differentiating between non-epileptic paroxysmal events and electrographic seizures is difficult. Up to two-thirds of clinical paroxysmal events are either unrecognized or misinterpreted by experienced neonatal staff [12]. Chan et al. reported that only 20% of paroxysmal events in 64 neonates in a neonatal intensive care unit were diagnosed as neonatal seizures on a simultaneous video-EEG recording [21]. Murray et al. also reported that only 27% of the clinically paroxysmal events were confirmed to be neonatal seizures [17]. In the present study, continuous video-EEG monitoring excluded electrographic seizures in more than two-thirds of the monitored neonates who had paroxysmal events mimicking seizures. Therefore, monitoring can help to differentiate non-epileptic paroxysmal events and electrographic seizures in neonates with a high risk of encephalopathy.

Neonatal seizures should be treated early to reduce mortality and morbidity and to improve neurodevelopment outcomes. However, the current seizure management guidelines are to treat clinical seizures identified through direct observation, with or without EEG confirmation, and this may increase the neonates to expose the risks of possible harm, either by medication overuse or undertreatment of subclinical seizures [33]. In the present study, half of the neonates with clinically abnormal paroxysmal events which staff believed to be a clinical seizure had received an AED during the continuous video-EEG monitoring. Continuous video EEG monitoring finally led to a change in clinical management in two-fifths of these neonates. Therefore, monitoring allows for the accurate titration and discontinuation of medication.

### Limitations

This study has several limitations. First, routine clinical practice at our NICU is to initiate or adjust AED when clinical seizures are diagnosed. However, AED may terminate convulsive seizures while electrographic-only seizures persist (uncoupling) [34]. Therefore, EEG monitoring after AED therapy has the potential to detect electrographic-only seizures. Second, the timing of EEG interpretation was variable, which would influence treatment strategies. Third, because the impact of continuous video-EEG-related management on outcomes is unknown, further prospective studies are needed to clarify this issue.

## 5. Conclusions

In this study, electrographic seizures were detected in one-third of the neonates with a high risk of encephalopathy, and these seizures were often electrographic-only. In addition, continuous video-EEG monitoring excluded electrographic seizures in two-thirds of the monitored neonates who had paroxysmal events mimicking seizures and led to a change in clinical management in 39.4% of the neonates. These findings suggest the routine use of continuous video-EEG in neonates with a high risk of encephalopathy.

## Data Availability

The data presented in this study are available on request from the corresponding author. The data are not publicly available due to privacy and ethical issue.

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
