# Peer review of "Electrographic Seizures in Neonates with a High Risk of Encephalopathy"

_children, 2022, doi:10.3390/children9060770_

Round 1

Reviewer 1 Report

Chen et al. manuscript entitled "Electrographic Seizures in Neonates with a High Risk of Encephalopathy" is an interesting article from the public health aspect, in which they utilized a video EEG monitoring approach in neonates that are at risk of encephalopathy. In particular, the strength of the article is that this prospective cohort study assesses continuously signal from the patients who are at high risk of developing seizures. In this way, the authors tap the timings of seizure initiation, propagation, and termination focussing on the encephalopathy group. The author chose this approach in order to characterize the type of seizure for further diagnosis. However, there is also weakness in the article as below. Line 27: I would suggest the authors start the background with the definition of encephalopathy. What makes encephalopathy neonates prone to developing status epilepticus in neonates? What is unknown about it in the field? Type of seizure characterization within neonates can still fall under this. What makes video EEG a perfect fit to tackle such a problem? Since you proceed forward to mention paroxysmal wording in line 41. I would suggest repeating this word in the introduction as well. In this way, readers would not get confused suddenly or jumped out of logical coherent sentences. Line 30-31: You can shorten the text January 2017, and December 2020 by something like "4-years". You can introduce the age range of the subject. You can explain more about how EEG was administered or what features this EEG had to sustain long-term recording as compared to earlier. All other unnecessary information you may provide in the relevant section of the manuscript later on. Authors should be able to mention the operating of EEG signals such as storing, processing, and analyzing in the Method sections. Line 32-42: The presentation style or wording style for reporting results should be improved. A more generic explanation sometimes makes more sense. Since authors have reported the results in %. The results will be stronger if authors can compare one group versus another by providing sample size N, p-Values, and the statistical tests used. In addition, I would define in the methods section how total N will be subdivided, what parameters are the study measuring in preceding results. All these logical flow is very crucial to make a logical flow for abstract. Line 36:I would use the full form "IQR" if is used for the first time. Overall, the abstraction sections need to rearrange the text, improve the result presentation section here, make it more contextual or logical. Line 46-66: This introduction seems more logical. Please make your logical flow consistent with the "Abstract" sections. Line 47-66: Introduction section: This must be prolonged. There are some missing gaps between sentences that authors need to address. For example how seizure is related to paroxysmal events. You need to expand by incorporating such gaps (See Pathak et al 2009, PMID: 19893074). Authors may also compare neonatal seizures versus adult or old age seizures and provide those data. As it will serve to give a wider view of the subject matter. You may also provide the common sign and symptoms of all applicable seizures types found in infants with your particular research design (here encephalopathy related) and a way to separate them with common types of seizures. Authors also need to expand the horizon of research toward a more molecular level or cellular level. I would suggest explaining something about "epileptic focus", spread, or propagation mechanism. Breaking down your cellular network epileptic activity to a single level help to understand many shortcomings. Recent findings from (Guan et al, 2018; PMID: 29641306) have reported how blocking of Kv2 channels can help in converting single action potential toward bursting. You may look at other papers from the Yaari laboratory for other channels related to epilepsy development (Chen et al. 2014, PMID 24920626 ) In this way, your introduction would be expanded and improved. Line 87-88: The date of approval might not be necessary here. However, if consent from patients has been taken or notified that the data will be used for publication, that will be relevant to mention. This sentence can be actually moved to line 71. Line 90-96: The description needs more elaboration. Which company EEG device was used, brand, the way of operation, where the patient was placed, how was the status of patient (fasting, drinking, food habits, or any other diet involved), medical history should be explained. All these details can be helpful here. Line 120: "ANE" please provide the full form if it is used for the first time. Please explain why 79-infants were excluded. Line 118-133: Give sentences that you wanted here to observe or find out using this technology as an opening sentence? You may explain how many groups or study subjects or parameters are designed for this approach (N, p-values, the test used will again be relevant before you start explaining your results). This result data can be easily structured in a pie-chart or appropriate bar diagram to make it more reflective or presentation friendly. Such a figure will help to simplify this text. You may then interpret those comparisons and rationale. Line 134-Line 151: Similar to Line 118-133. Use the same strategy for the presentation of those quantifications of results. Readers will always love to see your technology, for example, the sketch or photographs, how that machine looks like, how the children are doing while that instrumentation is in progress! If feasible, please provide such pictures or sketch at least (either in the Methodology section or results section). Line 161-line 212: Explain a few lines in your discussion about initiation, propagation, and termination of seizures, epileptic focus, and also ion channel perspective for seizure generation (Kv2 and KV7 and SK channels). In summary, the article has sufficient scientific content. There is more room for improvement in presentation style, the abstract section needs to be worked out, the date needs to be presented more in figure form, appropriate test (N, P, statistical approach) should be mentioned wherever significant result is reported, adding few perspectives of ion channel field in epilepsy generation at the cellular level will be helpful.

Reviewer 2 Report

See attached 

Round 2

Reviewer 2 Report

There are substantial improvements in the manuscript. However, with revision incorporated, a few additional issues arise and need to be addressed. 

  • The videos provided in the supplementary are raw. Mark the events with arrows or some other markers so that changes in EEG and behavior can be clearly understood.
  • Screenshots of EEG changes in a certain behavior (properly labeled) will give a better picture of changes.
  • Scheme-1 does not make logic. What exactly author want to show with this clumsy picture. Can be omitted. 
  • Scheme-2: need improvement. there are no error bars, statistical comparisons are missing. Accordingly, update legends with all the above information. 
